# Bioengineering Strategies for Corneal Endothelial Cell Injection Therapy: Advances, Challenges, and Clinical Translation

**DOI:** 10.3390/bioengineering12111162

**Published:** 2025-10-26

**Authors:** Yura Choi, Mi-Young Jung, Eunsun Han, Choul Yong Park

**Affiliations:** Department of Ophthalmology, Samsung Medical Center, Sungkyunkwan University School of Medicine, 81, Irwon-ro, Gangnam-gu, Seoul 06351, Republic of Korea; ychoi.op@gmail.com (Y.C.); myjung202@gmail.com (M.-Y.J.); ravw38@gmail.com (E.H.)

**Keywords:** cornea, endothelial cell, engineering, regeneration, injection therapy

## Abstract

Corneal endothelial dysfunction is a leading cause of vision impairment globally, traditionally managed through donor-dependent keratoplasty procedures. However, limitations in donor tissue availability, surgical complexity, and long-term graft survival have prompted the development of cell-based regenerative therapies. Among these, corneal endothelial cells (CECs) injection therapy has emerged as a minimally invasive alternative, offering the potential to restore endothelial function. This review provides a comprehensive analysis of recent advances in bioengineering strategies for CECs therapy, including cell sourcing from donor tissue, pluripotent stem cells, and transdifferentiated somatic cells; optimization of culture conditions and substrates; and delivery protocols that enhance cell adhesion and survival. We further examine clinical trial outcomes and propose future directions for clinical translation. The convergence of cell biology, biomaterials engineering, and translational medicine positions CECs injection therapy as a transformative solution to corneal blindness.

## 1. Introduction

Corneal transparency is essential for vision and is maintained by the corneal endothelium, a monolayer of hexagonal cells lining the posterior surface of the cornea [1]. These cells regulate stromal hydration through active ion transport and barrier functions [2,3]. Unlike other corneal layers, human corneal endothelial cells (CECs) exhibit limited proliferative capacity in vivo, rendering them vulnerable to age-related degeneration, trauma, and diseases such as Fuchs endothelial corneal dystrophy (FECD) and pseudophakic bullous keratopathy [1,4,5]. Loss of endothelial function leads to corneal edema, opacification, and progressive visual decline [4].

Current clinical management relies on keratoplasty procedures, including penetrating keratoplasty (PK), Descemet stripping automated endothelial keratoplasty (DSAEK), and Descemet membrane endothelial keratoplasty (DMEK) [4,6]. While these techniques have evolved to improve visual outcomes and reduce complications, they remain constrained by donor tissue shortages [7,8]. The global disparity in donor availability—particularly in low-resource settings—underscores the urgent need for alternative therapeutic strategies [7,9,10]. In a global survey by Gain et al. in 2016, global demand for keratoplasty is estimated at 12.7 million cases, with India and China accounting for approximately 7 million and 2 million cases, respectively. Despite this immense need, there is a significant global shortage of corneal graft tissue [10].

Cell injection therapy represents a paradigm shift in corneal endothelial regeneration [11,12]. By delivering cultured CECs directly into the anterior chamber, this approach aims to repopulate the endothelial layer and restore its physiological function without the need for full-thickness grafts. The success of this therapy depends on multiple factors: sourcing viable and expandable cells, optimizing culture conditions to preserve phenotype and function, and developing delivery protocols that ensure cell adhesion and integration [13].

Recent advances in bioengineering have addressed many of these challenges. The use of Rho-associated kinase (ROCK) inhibitors has significantly improved CECs proliferation and adhesion [14]. Substrate engineering, hypoxic culture conditions, and extracellular matrix (ECM) coatings have further enhanced cell viability and functionality [1,15,16]. Moreover, the exploration of alternative cell sources—including induced pluripotent stem cells (iPSCs), embryonic stem cells (ESCs), and transdifferentiated somatic cells—offers scalable solutions to overcome donor limitations [17,18,19,20,21,22,23].

This review summarizes the current state of CECs bioengineering for cell injection therapy, highlighting key innovations, clinical milestones, and unresolved challenges. We aim to provide a roadmap for researchers and clinicians seeking to advance this promising modality toward widespread clinical adoption.

## 2. Corneal Endothelial Cells Dysfunction and Current Treatments

CECs are a single layer of hexagonal cells lining the inner surface of the cornea, responsible for maintaining corneal transparency by regulating fluid balance between the stroma and aqueous humor [4]. These cells perform a dual function: they act as a semi-permeable barrier and actively pump ions to prevent stromal swelling [4]. This delicate equilibrium is essential for optical clarity and visual acuity. However, unlike epithelial cells, human CECs are known to be arrested in the G1 phase of the cell cycle and do not divide in vivo due to contact inhibition [5]. Loss of CECs is compensated by the migration of adjacent cells to fill the gap [5]. As corneal endothelial disorders progress, the normally uniform monolayer of cells becomes a heterogeneous layer with varying cell sizes and shapes [4,24]. This phenomenon is referred to as polymegathism and polymorphism of CECs. However, under laboratory conditions, cell division can be induced, allowing for serial cultivation [16].

When CECs density falls below a critical threshold—typically around 500 cells/mm^2^—the cornea loses its ability to maintain deturgescence [24]. The result is progressive stromal edema, epithelial bullae, and visual deterioration [4]. Among the most common causes of endothelial dysfunction are Fuchs endothelial corneal dystrophy (FECD), pseudophakic bullous keratopathy, and surgical trauma [4,25]. FECD, a genetically influenced degenerative condition, is characterized by the formation of guttae and progressive CECs loss, often manifesting in middle age and advancing with time [26]. Pseudophakic bullous keratopathy, on the other hand, arises from CECs damage during intraocular surgery, particularly cataract extraction, and remains a leading indication for corneal transplantation in many countries [27].

For decades, the primary treatment for CECs failure has been corneal transplantation. Penetrating keratoplasty (PK), the oldest technique, involves full-thickness replacement of the cornea and has historically yielded good anatomical outcomes [28]. However, it is associated with significant drawbacks, including high astigmatism, prolonged visual rehabilitation, and elevated risk of graft rejection [8]. More recently, lamellar techniques such as Descemet stripping automated endothelial keratoplasty (DSAEK) and Descemet membrane endothelial keratoplasty (DMEK) have revolutionized the field. These procedures selectively replace the diseased endothelium while preserving the anterior corneal architecture, resulting in faster recovery and improved visual outcomes [4,6]. A recent review by Shimizu et al. provided a comprehensive comparison of the clinical outcomes of PK, DSAEK and DMEK [6].

DMEK, in particular, has gained prominence due to its ability to restore near-normal vision and its low immunologic rejection rate [29,30]. However, despite its advantages, DMEK remains technically challenging and heavily reliant on high-quality donor tissue [30]. The global shortage of donor corneas continues to be a major obstacle, with estimates suggesting that only one cornea is available for every seventy patients in need worldwide [10]. This disparity is especially pronounced in low- and middle-income countries, where eye banking infrastructure is limited and surgical expertise may be scarce.

Moreover, even when donor tissue is available, long-term graft survival is not guaranteed [8,28,29]. Chronic CECs loss, immune-mediated rejection, and postoperative complications can compromise graft function over time [8]. These limitations have prompted researchers to explore alternative strategies that do not depend on donor tissue and offer more scalable, less invasive solutions.

Cell injection therapy has emerged as one such approach, aiming to restore endothelial function by delivering cultured CECs directly into the anterior chamber [11,12]. This technique bypasses the need for full-thickness grafts and leverages advances in cell culture, pharmacologic modulation, and bioengineering to promote cell adhesion and integration. Early clinical trials have demonstrated promising outcomes, suggesting that cell-based therapies may soon complement or even replace traditional keratoplasty in selected cases [11,31,32].

## 3. Corneal Endothelial Cells (CECs) Culture

Reliable isolation and expansion of healthy donor CECs is essential for successful cell injection therapy. Various factors required for optimal cultivation of CECs are illustrated in Figure 1. These cells must preserve their endothelial phenotype and pump function during ex vivo culture and, after transplantation, survive long-term in vivo while consistently maintaining endothelial-specific properties.

Culturing CECs is notoriously tricky due to their limited proliferative capacity and sensitivity to environmental conditions [5,16]. To support their growth and maintain their phenotype, researchers use specialized media supplemented with various key components [1,15,16]. In CECs culture, several key supplements are commonly used to support growth and maintain cellular phenotype. Fetal bovine serum (FBS), typically added at 5–10%, supplies essential growth factors, hormones, and nutrients that promote cell attachment and proliferation [1,5,15,33]. For clinical applications, human serum is often preferred to reduce xenogeneic risks and enhance transplantation compatibility [34]. Fibroblast growth factor (FGF), particularly basic FGF (bFGF), stimulates proliferation and is commonly used in low-serum conditions to support clonal expansion, but it also increases the risk of endothelial-to-mesenchymal transition (EnMT) [35,36].

Dual-media systems—alternating between a proliferative phase medium containing both FGF and serum, and a stabilization phase medium with reduced serum—are designed to preserve hexagonal cell morphology and inhibit EnMT [15,37]. Additionally, ECM coatings such as collagen IV, laminin, and fibronectin, are vital for cell attachment and structural integrity, with ECM derived from bovine CECs sometimes serving as a substitute for FGF in certain protocols [38,39,40].

Various researchers reported somewhat different formulations for the principal components of culture media used for CECs. Researchers use varying culture medium compositions because each laboratory has developed its own formulation through trial and error, aiming to optimize CEC proliferation while preserving their native phenotype. The most commonly used culture media for CECs are Ham’s F12/M199, Opti-MEM, and Human endothelial-SFM. Parekh et al. used Ham’s F12/M199 (1:1) supplemented with 5% FBS, 10 ng/mL bFGF, and 10 μM Y27632(ROCK inhibitor) [1]. Peh et al. developed a dual-media approach for culturing CECs, enabling initial expansion using proliferation media followed by maintenance with stabilization media [15]. In this study, proliferation media contained Ham’s F12 and M199 media as a base and supplements such as 5% FBS, 1% ITS (insulin, transferrin, selenium), 20 μg/mL amino acid, 10 ng/mL bFGF, and 10 μM Y27632(ROCK inhibitor). The maintenance media used human endothelial SFM as a base medium with supplements such as 5% FBS. Isolated CECs were kept in maintenance media overnight, then after attachment, medium was changed with proliferation media up to 14 days. Once CECs reached approximately 80% confluence, the culture medium was switched to maintenance media to promote stabilization. With this method, they reported excellent hexagonal morphology and expression of CECs markers. Okumura et al. used Opti-MEM-I for CECs culture and expansion. Their media contains 8% FBS, 5 ng/mL EGF, 20 μg/mL of amino acid, 200 mg/L calcium chloride, 0.08% chondroitin sulfate, and 10 μM Y27632 (ROCK inhibitor) [33].

### 3.1. Popular Culture Supplement for Corneal Endothelial Cells (CECs)

Various supplements and additives—including pituitary extract, transferrin, ascorbic acid, calcium chloride, and sodium selenite—have been investigated for their potential to enhance CECs culture [1,41] (Table 1). Among these, one of the most effective strategies for promoting CECs proliferation and survival has been the use of Y-27632, a ROCK inhibitor that blocks apoptotic pathways [14,42]. Rho-ROCK signaling orchestrates a diverse array of cellular functions—including cell adhesion, morphogenesis, migration, and cell cycle progression—by regulating cytoskeletal dynamics [42,43].

The use of L-ascorbate 2-phosphate (A-2P), an antioxidant that reduces oxidative stress, along with TGF-β inhibitors to suppress EnMT, has also been evaluated for enhancing the expansion of primary CECs [44,45]. Shima et al. demonstrated that A-2P and FGF-2 can enhance human CECs proliferation in vitro [45]. However, their study also revealed that FGF-2 may induce EnMT, characterized by loss of cell polarity and acquisition of a fibroblastic phenotype. This transition was associated with the secretion of type I collagen instead of type IV, contributing to retrocorneal membrane formation.

Additional approaches include the use of human serum and conditioned media, although these introduce variability due to inconsistencies in serum composition [34]. Human serum offers a rich source of growth factors and adhesion molecules, but its variability and potential for pathogen transmission limit reproducibility and scalability [46]. To address this, a new serum-free protocols have been explored as alternatives to fetal bovine serum [47]. Alonso-Alonso et al. used plasma rich in growth factors (PRGF-Endoret^®^, BTI, Vitoria, Spain) for CECs culture. They reported PRGF-expanded CECs share 46.9% of the gene expression profile with native cornea endothelium and express all studied corneal endothelial marker [47].

Human platelet lysate (HPL) was suggested as a replacement for human serum or fetal bovine serum (FBS) for CECs culture. (Table 2) [48,49,50,51]. As cell injection therapy becomes increasingly widespread, the use of human-derived components is expected to rise in order to minimize the risks associated with animal-derived substances, in alignment with regulatory guidelines. Treatment of immortalized CECs with the HPL increased their viability, enhanced the wound closure rate, and maintained cell growth and typical hexagonal morphology [49]. HPL is rich in fibrinogen, PDGF, hEGF, VEGF and bFGF which promote CECs attachment and proliferation. In the study by Petsoglou et al., 5% HPL significantly increased proliferation and viability of primary culture of CECs, and immunostaining indicated that HPL increased ZO-1 and CD166 expression but not Na^+^/K^+^-ATPase [48]. Mishan et al. also reported a higher expression of Na^+^/K^+^-ATPase and ZO-1 in the culture of CECs treated with 20% HPL as compared with FBS [50]. Talpan et al. found that both 2% and 5% HPL led to the upregulation of cytoprotective, anti-inflammatory and anti-fibrotic genes while downregulating pro-inflammatory/apoptotic gene compared to FBS supplement [52].

**Table 1 bioengineering-12-01162-t001:** Culture supplement for corneal endothelial cell.

Supplement	Common Dose Ranges	Function/Benefit	Reported Efficacy
Fetal Bovine Serum (FBS)	2–10% (*v/v*); 2–5% for maintenance, 5–10% for proliferation	Provides growth factors and nutrients	Widely used; supports proliferation but may induce fibroblastic changes at high doses [41]
Human Serum	2–10% (*v/v*); typically 5%	Alternative to FBS; reduces xenogenic risks	Promotes proliferation with better morphology preservation [34]
bFGF (Basic Fibroblast Growth Factor)	5–20 ng/mL (commonly 10 ng/mL)	Stimulates proliferation and survival	Essential for clonal expansion; improves cell yield and risk of EnMT [35,36]
EGF (Epidermal Growth Factor)	5–20 ng/mL (commonly 10 ng/mL)	Enhances proliferation and migration	Supports wound healing and cell expansion; used in combination with other factors [41,53]
ROCK Inhibitor (Y-27632)	3–10 μM (10 μM initially, 5 μM for maintenance)	Inhibits apoptosis and promotes cell adhesion	Dramatically improves survival and proliferation; widely adopted in recent protocols [42,43]
Ascorbic Acid (Vitamin C)	50–200 μM (commonly 100 μM)	Antioxidant; supports collagen synthesis	Enhances cell viability and reduces oxidative stress [45]
Insulin-Transferrin-Selenium (ITS)	0.5×–1× (typically 1×)	Supports metabolic activity and cell growth	Improves proliferation and maintains endothelial phenotype [54]
Chondroitin Sulfate	0.08–0.2% (*w/v*)	Maintains cell shape and barrier function	Used in organ culture; helps preserve native morphology [41]
Stem cell conditioned media	10–50% (*v/v*)	contains growth factors, metabolites and ECM proteins secreted by the cells	Maintains stemness and improves regenerative potential in some protocols [55,56]
Extracellular Matrix Coatings	Collagen IV (10–50 μg/mL), Laminin (5–20 μg/mL), Fibronectin (2–10 μg/mL)	Collagen IV, VIII, laminin, fibronectin	Critical for CECs adhesion and morphology [57]
Nerve growth factor (NGF)	10–100 ng/mL	may reduce apoptosis and promote regenerative signaling	Enhance CEC proliferation and survival [58,59]
Bovine pituitary extract	30–100 μg/mL	rich source of growth factors (FGF, EGF, etc.)	helps cells overcome senescence and maintain a healthy monolayer [41].

**Table 2 bioengineering-12-01162-t002:** Comparison of human serum and human platelet lysate for cell culture.

Attribute	Human Serum	Human Platelet Lysate (HPL)
Source	Serum separated from donated whole blood after clotting	Lysate made from pooled human platelets by freeze–thaw, sonication, or other lysis
Key components	Albumin, immunoglobulins, complement proteins, low–moderate growth factor levels	High concentrations of growth factors and cytokines (PDGF; EGF; VEGF; TGF-β)
Proliferation support	Moderate cell proliferation support	Strong proliferation support for many cell types, notably MSCs
Batch-to-batch variability	Donor-dependent variability; pooling reduces variability	Donor- and process-dependent variability; pooling and standardized production reduce variability
Clinical translation	Used in clinical protocols; requires rigorous screening and processing	Widely adopted for clinical-grade expansion; many GMP-produced HPL products available
Preparation/processing	Clotting, centrifugation, heat-inactivation commonly required	Platelet collection, lysis (freeze–thaw or mechanical), centrifugation, optional heparin to prevent fibrin gelation
Immunogenicity/xenogeneic risk	Human-derived, low xenogeneic risk	Human-derived, avoids animal components and xenogeneic risk
Effect on cell phenotype & function	Maintains phenotype; growth factor levels lower than HPL	Maintains phenotype and immunomodulatory functions; can increase VEGF secretion and proliferation
Cryopreservation performance	Suitable but sometimes inferior to platelet-derived supplements	Comparable or superior cryopreservation outcomes in some studies
Cost & availability	Moderate cost; depends on donor supply	Variable cost; scalable with pooled platelet plasma and commercial GMP HPL products

### 3.2. Donor Effect on Cornea Endothelial Cell Culture

The proliferation of CECs is dependent on donor conditions [60] (Table 3). Healthy donor CECs are easy to culture and maintain their phenotype well even after multiple passages. CECs harvested from young and healthy donors are more proliferative with phenotype preservation. Non-traumatic causes (e.g., natural causes) linked with better CECs proliferation. CECs from eyes with previous intraocular surgery showed decreased proliferation with easy change in phenotypes.

### 3.3. Signal Pathways to Maintain CECs Phenotype

Previous studies have shown that the homeostasis and phenotype maintenance of CECs are governed by complex signaling pathways. (Table 4) Understanding these pathways will provide a solid foundation for advancing CECs culture technique in the future. Figure 2 illustrates the various signaling pathways involved in successful culture of CECs.

TGF-β signal pathway is closely implicated with CECs [68]. The binding of TGF-β to its receptors initiates the activation of receptor-regulated Smads (R-Smads), primarily Smad 2 and Smad 3. These R-Smads form a complex with the common-mediator Smad (Co-Smad), Smad 4, which then translocates into the nucleus to regulate the transcription of target genes. Smads involved in the TGF-β signaling pathway are categorized into three groups: receptor-associated Smads (Smad 2, Smad 3), the co-mediator Smad (Smad 4), and inhibitory Smads (Smad 7). Funaki et al. demonstrated that overexpression of Smad 7 suppresses the TGF-β–mediated inhibition of cell proliferation by blocking Smad 2 phosphorylation [69]. Furthermore, Smad 7-transfected corneas showed reduced cell density loss and enhanced wound healing, whereas Smad 3 overexpression led to decreased cell density and delayed wound repair [70]. The mechanism of EnMT in CECs involves the TGF-β signaling pathway, and one of the important inhibitors of the TGF-β/Smad 2/3 pathway is sirtuin-1 (SIRT1). Yu et al. reported resveratrol activation of SIRT1 downregulated the expression levels of alpha smooth muscle actin (α-SMA), vimentin, and Snail, while upregulated the expression levels of E-cadherin and Na^+^/K^+^-ATPase [71]. Ryu et al. treated CECs with AMF30a, peptidylarginine deiminase 2 (PAD2) inhibitor, and found AMF30a promoted the proliferation and protected against TGF-β-induced senescence in human CECs [72]. Joko et al. reported that addition of SB431542, an inhibitor of TGF-β could suppress EnMT [73].

The Notch, Wnt, and Hippo signaling pathways are evolutionarily conserved systems that play essential roles in regulating cell behavior, particularly in tissue development, regeneration, and disease progression [74]. Each pathway contributes to cell maintenance, differentiation, and proliferation, and is vital for maintaining tissue homeostasis and controlling organ size. Despite their distinct molecular components, these pathways share overlapping functions and often interact to fine-tune cellular outcomes [74,75]. For instance, Hippo-YAP/TAZ signaling can either suppress or enhance Wnt/β-catenin activity depending on the context, while Notch and Hippo pathways form feedback loops that influence organ growth and repair. Moreover, Wnt and Notch signaling coordinate stem cell lineage decisions, especially in epithelial tissues, underscoring their integrated roles in maintaining physiological balance [76].

Hirato-Tominaga et al. identified LGR5, a Wnt target, uniquely expressed in the peripheral region of CECs and that LGR5(+) cells have some stem/progenitor cell characteristics. LGR5 expression maintained endothelial cell phenotypes and inhibited mesenchymal transformation through the Wnt pathway [77]. Additionally, R-spondin1 promoted proliferation in rabbit and human CECs through Wnt/β-catenin signaling [33]. Wnt/β-catenin was found to suppress contact inhibition in EDTA-bFGF–treated cultures, though this method risked inducing EnMT [78]. Another study revealed that IL-1–induced Wnt5a enhances CECs migration via a β-catenin–independent Wnt/Fzd pathway involving Cdc42 activation and RhoA inhibition [79]. Maurizi et al. conducted proteomics analysis and found that β-catenin activation was necessary during rabbit CECs proliferation and both pro-proliferative activity of basic fibroblast growth factor and anti-proliferative effects of TGF-β were regulated through β-catenin [80].

Immunofluorescence studies showed that Notch and its ligand Delta are present in the corneal epithelium but absent in CECs [81]. Despite this, Notch signaling is known to interact with pathways like Wnt and TGF-β/Smad, suggesting its potential role in CECs proliferation [82]. Notch was also identified as a downstream effector of TGF-β signaling, contributing to injury-induced CECs fibrosis [82]. Recent research in rat CECs revealed that Notch activation suppresses proliferation, while inhibition of Notch with DAPT restores the parental CECs phenotype [83]. Catala et al. found that Notch and TGF-*β* pathways have increased activity in species with non-proliferative CECs, which might be associated with their low proliferation [84]. Figure 3 illustrates the crosstalk among key signaling pathways that regulate the phenotype of CECs, including Wnt/β-catenin, TGF-β/BMP-SMAD, Notch, and Hippo-YAP/TAZ pathways.

Rho-associated protein kinase (ROCK) is a serine/threonine kinase that regulates cellular shape and adhesion through its action on the cytoskeleton [85,86]. ROCK inhibitors, widely used in glaucoma treatment, have recently shown promise in corneal endothelial disease therapy by enhancing cell proliferation, adhesion, and reducing apoptosis [43]. The role of ROCK in CECs in clinical cases and laboratory experiment were thoroughly reviewed by Singh et al. and Futterknecht et al. [14,87]. Previous studies have demonstrated that inhibition of ROCK using various inhibitors—such as Y-27632, Ripasudil, Netarsudil, and Fasudil—enhances the adhesion, migration, and proliferation of cultured CECs. Furthermore, increased CEC densities and subsequent reversal of corneal edema have been reported in several pathological conditions, including Fuchs endothelial dystrophy, bullous keratopathy, and phacoemulsification-induced CEC damage [14,87].

**Table 4 bioengineering-12-01162-t004:** Important signaling pathways in corneal endothelial cell culture.

Pathway	Role in CECs	Effect on Phenotype	Modulation Strategy	Reference
TGF-β	Induce G1 arrest in vivo. Drives EnMT and fibrosis under stress or serum-rich conditions	Loss of polygonal shape, gain of fibroblastic traits	Inhibit with SB431542 or LY2109761	[68]
ROCK	Regulates cytoskeleton, adhesion, and survival	Supports proliferation and phenotype retention	Inhibit with Y27632 or Ripasudil	[14]
MAPK (ERK)	Promotes proliferation but may reduce cell density	Mixed effects; can reduce ECD	Use with caution or inhibit selectively	[88]
MAPK (p38)	Stress response and senescence regulation	Inhibition increases ECD and preserves phenotype	Inhibit with SB203580 or SB202190	[89]
BMP	Counters TGF-β-induced EnMT	Maintains endothelial morphology	Activate BMP-7	[90]
Notch	Regulates cell fate and EnMT, especially post-injury	Excess activation may cause phenotype loss	Inhibit with γ-secretase inhibitors	[83]
Wnt	Stimulates proliferation and regeneration	Promotes recovery and phenotype retention	Activate with Wnt mimetics or ligands	[33,77]

### 3.4. Cornea Endothelial Cell Phenotype Markers for Validation of Culture

Maintaining phenotypic fidelity during culture is essential for therapeutic efficacy. CECs must retain their barrier and pump functions, which are mediated by proteins such as ZO-1, Na^+^/K^+^-ATPase, and N-cadherin (Table 5) [1,91]. Immunocytochemistry and gene expression profiling are routinely used to confirm the presence of these markers, while functional assays—such as transepithelial electrical resistance (TEER) and fluid transport measurements—provide evidence of physiological competence.

CD166 helps identify and maintain the correct phenotype during ex vivo expansion of CECs [92]. It is especially useful in preventing EnMT, which can compromise graft quality. Cadherin-2 is essential for cell junction integrity. It ensures that the endothelial monolayer remains cohesive and functional, which is vital for corneal transparency and fluid regulation [18]. SLC4A11 is more about cell survival and metabolism [92]. Loss of SLC4A11 disrupts mitochondrial function and increases autophagy and mitophagy, which can lead to corneal endothelial dystrophies. An example of this is the panel developed and used by Kinoshita and colleagues [91]. They reported that CECs specific cell markers (e.g., CD166+, sPRDx-6+, CD44^−^, CD 105^−^, CD 26^−^, CD 24^−^) have helped to define the optimal cell type for cell injection therapy [93,94,95]. In this panel, CD166 was used as a marker for CECs and the negative makers were analyzed to exclude the fibroblastic-like phenotype.

For cell injection therapy in clinical trials, the final cultured cell product is of utmost importance. Above all, it must be safe and demonstrate reliable functionality. However, any primary cell culture process carries the risk of heterogeneity in the quality of the expanded cells. CECs cultures are particularly susceptible to issues such as EnMT and keratocyte contamination. Therefore, rigorous evaluation of the cultured cells cannot be overemphasized.

**Table 5 bioengineering-12-01162-t005:** Important makers for Corneal Endothelial Cells culture.

Marker	Full Name	Function in CECs	Research/Clinical Relevance	Reference
Na^+^/K^+^-ATPase	Sodium-Potassium ATPase	Maintains ionic balance and fluid transport across the endothelium	Gold-standard functional marker for CECs identity and pump function	[18]
ZO-1	Zonula Occludens-1	Tight junction protein that maintains barrier integrity	Used to assess monolayer integrity and hexagonal morphology	[18]
CD166	Activated Leukocyte Cell Adhesion Molecule	Cell adhesion and phenotype maintenance	Marker of non-fibroblastic, functional CECs	[92]
sPrdx6	secreted form of Peroxiredoxin 6	neutralize reactive oxygen species	Protect CECs from oxidative damage and important marker of functional CECs	[92,96]
CD73	Ecto-5′-nucleotidase	Associated with fibroblastic transformation	High expression indicates mesenchymal-like phenotype; used for negative selection	[91,97]
CD44	Hyaluronan receptor	Cell adhesion and migration	Elevated in fibroblastic CECs; linked to EnMT	[91]
CD49e	Integrin α5	ECM interaction and cell adhesion	Marker of fibroblastic phenotype	[91]
CD98	4F2 cell-surface antigen heavy chain	Amino acid transport and cell activation	Elevated in functional, non-fibroblastic CECs	[91]
CD340	HER2/neu	Growth factor receptor	Associated with non-fibroblastic phenotype	[91]
N-Cadherin	Cadherin-2	Cell–cell adhesion and junction stability	Maintains endothelial monolayer structure	[18]
SLC4A11	Solute Carrier Family 4 Member 11	Ion transport and mitochondrial homeostasis	Mutations linked to endothelial dystrophies; essential for cell survival	[92]
Collagen I	Type I Collagen	ECM component	Overexpression indicates fibroblastic transformation	[98]
Fibronectin	ECM glycoprotein	Cell adhesion and migration	Marker of mesenchymal phenotype; elevated during EnMT	[98]

### 3.5. Evaluating CECs Functionality

It is critical to verify that cultured CECs possess active ion transport mechanisms, which underline their fluid pump function to maintain corneal deturgence. Although markers like Na^+^/K^+^-ATPase and SLC4A4 are commonly used to indicate transporter presence, their expression alone does not confirm functional activity. As a result, researchers have developed a range of in vitro, ex vivo, and in vivo methods to more accurately assess CECs performance [99,100,101]. Among these, measuring ion movement across a monolayer offers a practical way to evaluate transport activity. Conventional techniques such as transepithelial electrical resistance (TEER) provide insight into barrier integrity [99,102]. Ussing chamber analysis allows for direct ion flux measurement, though it requires careful control of environmental variables to ensure consistency [103,104]. Alternatively, ex vivo anterior chamber infusion models were also introduced [101]. Detailed protocols for TEER and Ussing chamber is available at the manufacturers’ websites (www.harvardapparatus.com/media/harvard/pdf/Ussing_Chamber_Systems.pdf, accessed on 6 October 2025)

Monitoring corneal thickness changes offers a useful proxy for pump activity, and bioreactor systems have been introduced to better replicate the ocular environment for dynamic testing. While no single method is definitive, combining multiple approaches enhances reliability. To streamline quality control, emerging technologies like organ-on-a-chip platforms—designed to mimic organ-level physiology—hold promise for developing scalable, reproducible models of the corneal endothelial barrier [102]. Organ-on-a-chip technology represents a significant advancement in biomedical research, designed to address the shortcomings of traditional in vitro and in vivo models [105]. This technology has been extensively studied for drug toxicity assessment using cultured corneal epithelial cells; however, there are currently no documented applications involving corneal endothelial cells in the available literature. These systems could play a pivotal role in validating each batch of CECs prior to clinical application.

## 4. Bioengineering for Corneal Endothelial Cell

Compared with conventional corneal endothelial transplantation, in vitro expansion of CECs for cell injection therapy allows cells harvested from a single donor to be amplified under controlled laboratory conditions, and it enables the use of diverse bioengineering strategies to enhance cell function and improve adhesion to the posterior corneal surface after injection. Together, these bioengineering strategies have enabled the expansion of CECs from older donors, improved consistency across batches, and enhanced the likelihood of successful engraftment following injection. As protocols continue to evolve, the integration of biomaterials and microfluidic systems may further refine the process, paving the way for scalable and standardized production of therapeutic-grade CECs [106,107].

### 4.1. Enhancing CECs Viability

Oxygen tension is another key variable in CECs culture. While most in vitro systems operate under atmospheric oxygen (~18%), the physiological environment of the anterior chamber is relatively hypoxic. The physioxic partial pressure of O_2_ at the endothelial surface of the central human cornea is approximately 21 mm Hg (~2.8%) [108]. This is substantially lower than the ~18% O_2_ that is present in most ambient air tissue culture incubators. Culturing CECs under reduced oxygen levels (2.5%) has been associated with decreased oxidative stress, enhanced mitochondrial function, and prolonged cell viability [109,110]. NOX4, an isoform of NADPH oxidase, contributes to CECs apoptosis by generating excessive ROS, which disrupts cellular homeostasis. Elevated NOX4 levels activate oxidative stress pathways, including endoplasmic reticulum (ER) stress, impairing protein regulation and CECs function [111]. These findings suggest that hypoxic conditions may better support the metabolic demands of CECs and reduce senescence during expansion.

The proliferation of CECs can potentially be enhanced through the regulation of gene expression involved in cell cycle–related signaling pathways, transferring antiapoptotic genes, modulation of protein-coding genes associated with contact inhibition, and targeting of known disease-related genes implicated in CECs dysfunction. (Table 6) Transcription factor 4 (TCF4) mutation is related to FECD leading to CECs death with aging [112]. Yan et al. developed TCF4 overexpressing HCECs cell lines and reported increased CECs migration potential [113]. Fuchsluger et al. used plasmid vector and transferred p35 and Bcl-x_L_ genes to cultured CECs. Overexpression of p35 and Bcl-x_L_ was protective against apoptosis induced by extrinsic (death receptor) or intrinsic (mitochondrial) apoptotic pathways [114].

The cell cycle is a tightly regulated process common to all eukaryotic cells, primarily controlled by cyclin-dependent kinases (CDKs) and their interactions with cyclins [115,116]. Upon mitogenic stimulation, cyclins activate CDKs, leading to phosphorylation of retinoblastoma (Rb) proteins and release of E2F transcription factors, which drive cell cycle progression. This activation is reinforced by positive feedback, pushing the cell past the restriction point into continued division [117]. Negative regulation is mediated by cyclin-dependent kinase inhibitors (CKIs), including the INK4 and CIP/KIP families, which block CDK activity and maintain G0 or induce G1 arrest [117]. Additionally, the tumor suppressor p53 can halt the cycle or trigger apoptosis in response to DNA damage, primarily through induction of p21CIP1, which inhibits cyclin-CDK complexes and E2F activation [118]. Strategies to enhance the proliferation of CECs may involve targeting key regulatory proteins that mediate both positive and negative signaling pathways. By modulating these molecular checkpoints, it is possible to promote controlled cellular expansion while preserving phenotypic stability.

**Table 6 bioengineering-12-01162-t006:** Targets for genetic modulation for CECs proliferation and results.

Targets	Results
E2F2	Transduction with E2F2 resulted in progression from the G1 to the S phase and increases CECs density [119].
ZONAB	Modulation of contact inhibition by ZO-1/ZONAB gene transfer increases CECs density [120].
ZO-1	ZO-1 downregulation using ZO-1 shRNA increases CECs proliferation on donor grafts [120].
CKIs and p53	Stable expression of p53 shRNA enhanced cell survival by approximately 12 population doublings. Combining p53 knockdown with TERT overexpression resulted in the immortalization of CECs [121]. Downregulation of p27KIP1 using siRNA led to a 30% increase in CECs density [122].
p120 Catenin/Kaiso	p120 catenin siRNA increases CECs proliferation by inhibiting Kaiso [123].
SOX2	SOX2 overexpression increases cell proliferation and viability in CECs [124].
SIRT1	SIRT1 overexpression increases CECs proliferation and viability [125].

Virus-derived oncogenes have demonstrated utility in enhancing CEC proliferation and generating immortalized cell lines such as HCEC-12 and B4G12. (Table 7) However, their application raises important safety concerns. Chief among these is the potential for tumorigenicity due to dysregulated cell cycle control and genomic integration [126]. These risks necessitate a careful risk-benefit analysis, particularly when considering clinical translation. As safer alternatives, emerging non-viral approaches—such as CRISPR-based gene editing—offer precise modulation of proliferation-related pathways without the integration risks associated with viral vectors [127]. These technologies represent a promising direction for expanding CECs while minimizing long-term safety concerns.

Human telomerase reverse transcriptase (TERT) is the catalytic subunit of telomerase, an enzyme that prevents telomere shortening—a process linked to cellular aging and apoptosis—by extending chromosome ends during cell division [134]. While telomerase is typically inactive in most differentiated somatic cells, its introduction can enhance proliferative capacity and potentially induce immortalization. In CECs, TERT alone had limited impact under standard conditions but extended cell survival under reduced oxidative stress and enabled a fast-proliferating subpopulation to reach 36 population doublings without signs of tumorigenicity [121]. These TERT-transfected cells retained key CECs features such as contact inhibition and protein markers [135]. When combined with oncogenes like HPV 16 E6 or CDK4, TERT successfully immortalized CECs, producing cell lines that maintained normal characteristics and closely resembled native CECs in transcriptome profiles [121].

Exosomes, the smallest subtype of extracellular vesicles, are lipid-bound nanovesicles (30–150 nm) secreted by most cell types [132]. Extracellular vesicles are categorized into apoptotic bodies, microvesicles, and exosomes based on their size and biogenesis. Owing to their role in intracellular communication and excellent biocompatibility, exosomes serve as effective delivery vehicles, transporting nucleic acids such as microRNAs, proteins, and lipids from their parent cells [136,137]. Leveraging these properties, exosomes can enhance CECs culture efficiency and improve post-transplantation survival. Exosomes derived from adipose-derived mesenchymal stem cells (ASCs) promoted CECs proliferation and attenuate TGF-β- or H_2_O_2_-induced oxidative stress, senescence, endothelial-mesenchymal transition, and mitophagy. In vivo, ASC-derived exosomes accelerate corneal endothelial wound healing and protect against cryoinjury-induced damage [138]. Moreover, miR-184 in aqueous humor prevented oxidative stress-induced degeneration of CECs, thereby contributing to tissue homeostasis [139]. miR-302a enhanced the regeneration of CECs by eliminating IFN-γ-induced senescence and ER stress, and promoted the CECs regeneration in rats after cryo-injury [140]. Inhibiting miR-195-5p induced the proliferation of CECs while preserving normal morphology [141].

### 4.2. Enhancing CECs Attachment to Recipient Cornea

Recent advances in nanotechnology have enabled the use of magnetic nanoparticles to improve CECs delivery and attachment, offering promising alternatives to conventional cell injection therapies for corneal endothelial dysfunction.

Xia et al. loaded with superparamagnetic nanoparticles, and injected into the anterior chamber of adult rabbits immediately after endothelial cell or Descemet’s membrane stripping [142]. Magnetic CECs integrated onto the recipient corneas with intact Descemet’s membrane [142]. Moysidis et al. explored the impact of 50 nm diameter magnetic nanoparticles on the delivery of cadaveric donor CECs, and found 2.4-fold increase in cell density compared to conventional gravity-driven methods [143]. Park et al. labeled primary CECs with commercially available magnetic micro- or nanoparticles [144]. The application of a magnetic field during cell culture successfully demonstrated that magnetic particle-loaded CECs moved toward the magnet area. Mimura et al. investigated the use of polyethylene glycol-coated magnetic nanoparticles to reduce aggregation and immune response, while maintaining magnetic responsiveness [145]. Their results confirmed enhanced cell localization and minimal inflammation.

Zhao et al. developed a convenient and injectable magnetic hyaluronic acid (HA) gel system by incorporating superparamagnetic nanoparticles, enabling efficient delivery CECs [146]. The resulting gel exhibits desirable shear-thinning behavior and superparamagnetic properties, while maintaining high cell viability. Furthermore, it demonstrates excellent capabilities in cell concentration and precise localization, making it a promising platform for targeted cell therapy.

The phase 1 clinical trial result of magnetic nanoparticle loaded CECs (EO2002, Emmecell, Menlo Park, CA, USA) was pressed in 2024. In the multicenter, randomized trial, a cohort receiving 150,000 endothelial cells achieved a mean gain of 11 letters in BCVA at 6 months, with 38% of patients achieving at least a 15-letter gain. No adverse reactions were reported. Nonetheless, the use of magnetic particles warrants continued caution due to unresolved concerns regarding potential cytotoxicity and long-term biocompatibility.

## 5. Alternative Source of CECs

Recent advances in stem cell biology have opened new avenues for generating CECs-like cells from pluripotent sources. Induced pluripotent stem cells (iPSCs) offer the promise of patient-specific therapies with reduced immunogenic risk [17]. Most protocols begin by directing iPSCs toward a neural crest cell (NCC) lineage using factors like BMP4, FGF2, and ROCK inhibitors. Chambers et al. introduced a widely adopted dual-SMAD inhibition method (SB431542 and LDN193189) for efficient NCC induction, producing cells that express SOX10, FOXD3, and PAX3 and can be further differentiated into endothelial-like cells [147,148]. Zhao et al. expanded this approach by incorporating Wnt/β-catenin activation, TGF-β inhibition, and ROCK modulation [149]. Hatou et al. bypassed the NCC stage using a xeno-free, chemically defined medium with FGF, EGF, insulin–transferrin–selenium-A, ascorbic acid, and Y-27632, achieving differentiation within 28 days and maintaining cell identity post-cryopreservation [18]. Grönroos et al. developed a stepwise, small-molecule-based protocol using SB431542, CHIR99021, and retinoic acid, later adapting it for 3D bioprinting with hyaluronic acid bioinks [150]. Other strategies include small-molecule screening to enhance marker expression and GSK-3β inhibition to preserve hexagonal morphology and pump function [151].

The therapeutic potential of iPSC-derived CECs like cells has been validated in animal models including monkeys [18,19]. In a first-in-human study, Hirayama et al. used clinical-grade iPSCs (QHJI01s04, CiRA, Kyoto University) cultured under xeno-free conditions and differentiated into endothelial-like cells with strong expression of Na^+^/K^+^-ATPase, ZO-1, N-cadherin, and PITX2 [32]. Flow cytometry confirmed over 97% marker expression, with undifferentiated cells below 0.01%. Following injection of 8 × 10^5^ cells into a patient with bullous keratopathy, corneal clarity and visual acuity improved without adverse events or tumor formation at one year. The first-in-human clinical study of an allogenic iPSC-derived corneal endothelial cell substitute transplantation was conducted in 73-year-old male patient with bullous keratopathy. Decrease in cornea thickness and improvement of visual acuity was observed for 12 months [32].

Song et al. used a modified two-stage differentiation method to convert H9 human embryonic stem cells (hESCs) to neural crest cells (NCCs), then further into CECs-like cells. The CECs-like cells was treated with bovine CECs conditional medium morphologically best resembled primary CECs and highly expressed Na^+^-K^+^-ATPase, AQP1, Col8a and ZO-1 [152]. McCabe et al. also CECs-like cells from hESCs. These cells were confirmed to express all major corneal endothelial pump transcripts, and microarray analysis showed a 96% similarity to adult human CECs [153]. Other studies have also reported successful generation of CECs-like cells from hESCs [23,154].

Several groups have investigated non-corneal cell sources for treating endothelial dysfunction. Vascular endothelial cells have been evaluated in tissue-engineered corneal transplantation and demonstrated partial improvement of corneal edema in experimental models. However, in the monkey model of bullous keratopathy, transplantation of these cells resulted in only limited restoration of corneal thickness and transparency, and was accompanied by immunological rejection [20,155]. Retinal pigment epithelial (RPE) cells, which share hexagonal morphology and express key markers such as ZO-1 and Na^+^/K^+^-ATPase, have also shown therapeutic potential. In rabbit models of corneal endothelial dysfunction, both primary and human embryonic stem cell hESC-derived RPE cells successfully restored corneal thickness and transparency. Intracameral injection of cultivated rabbit primary RPE cells, CECs, and hESC-derived RPE cells led to resolution of corneal edema and reduction in corneal thickness, with therapeutic effects comparable to those observed with CECs. Notably, hESC-derived RPE cell transplantation also restored corneal clarity and thickness within one month after the surgery. Further analysis revealed upregulation of CECs markers such as CD200 and S100A4, alongside significant downregulation of RPE-specific markers including OTX2, BEST1, and MITF in the transplanted RPE cells [21].

More recently, transdifferentiation strategies have emerged, aiming to convert accessible somatic cells directly into corneal cells such as corneal epithelial cells or CECs without passing through a pluripotent state [22,156]. Pan et al. developed a stepwise strategy to generate chemically induced neural crest cells and subsequently chemically induced CECs from mouse fibroblasts [156]. For differentiation, they used a medium containing SB431542 (TGF-β signaling inhibitor) and CKI-7 (Wnt signaling inhibitor). The resulting CECs-like cells exhibited strong proliferative capacity and formed a monolayer of hexagonal cells. These CECs-like cells expressed corneal endothelial cell-specific markers, confirming their identity. These findings highlight the translational potential of alternative cell sources—including vascular endothelial cells, RPE cells, and transdifferentiated somatic cells—as promising candidates for future corneal endothelial therapies.

Whether derived from donor tissue, reprogrammed stem cells, or transdifferentiated somatic cells, the therapeutic utility of CECs hinges on their ability to form a cohesive monolayer, maintain pump function, and integrate seamlessly into the host cornea. As the field advances, comparative studies and standardized characterization protocols will be essential to determine the optimal cell source for widespread clinical deployment. At the same time, ethical considerations and cost comparisons with donor-derived cells must be carefully addressed in the development of these alternative approaches.

## 6. Cell Delivery and Engraftment Enhancement

The therapeutic success of CECs injection therapy hinges not only on the quality of the cultured cells but also on the precision and efficacy of their delivery into the anterior chamber (Figure 4). Unlike traditional keratoplasty, which relies on surgical implantation of donor tissue, cell injection therapy demands a biologically favorable environment where injected cells can adhere to Descemet’s membrane, survive, and reconstitute a functional monolayer. Achieving this outcome requires careful orchestration of delivery parameters, pharmacologic support, and postoperative management. The injection procedure typically involves suspending cultured CECs in a biocompatible medium and introducing them into the anterior chamber through a small corneal incision. The volume and concentration of the cell suspension are calibrated to ensure adequate coverage of the posterior corneal surface without inducing intraocular pressure elevation [11]. One of the most critical factors influencing engraftment is the patient’s positioning following injection. Studies have shown that maintaining a face-down posture for several hours postoperatively facilitates gravitational settling of the cells onto Descemet’s membrane, promoting uniform adhesion and minimizing cell loss [11].

Pharmacologic modulation plays a central role in enhancing cell survival and integration after cell injection therapy. ROCK inhibitors, such as Y-27632, have emerged as indispensable adjuncts in both the culture and delivery phases [11,87]. These agents modulate actin cytoskeleton dynamics, reduce apoptosis, and promote tight junction formation, thereby improving the likelihood of successful engraftment [43,87]. In addition to pharmacologic support, mechanical and biochemical conditioning of the recipient corneal surface can further enhance cell adhesion. Gentle scraping of Descemet’s membrane to remove residual dysfunctional endothelium exposes native extracellular matrix components, creating a more receptive substrate for cell attachment [157]. Some researchers have explored micro-patterned surfaces to optimize adhesion, although these techniques remain largely experimental [158].

Emerging biomaterial-based strategies offer new possibilities for improving cell retention and spatial distribution. Thermoresponsive hydrogels, for example, can be injected in liquid form and solidify at body temperature, providing a scaffold that supports corneal endothelial cell attachment and prevents dispersion [159,160]. Similarly, biodegradable microparticles or carriers have been designed to release cells gradually enhancing engraftment efficiency and reducing the risk of clumping or uneven coverage [161]. Despite these mechanistic advantages, translational hurdles remain. These include ensuring consistent material-cell interactions, avoiding immunogenicity or toxicity, and scaling up production under GMP conditions.

Together, these delivery and engraftment strategies form the backbone of CECs injection therapy. As protocols continue to evolve, the integration of biomaterials, pharmacologic enhancers, and patient-specific adjustments will be key to optimizing outcomes and expanding the applicability of this minimally invasive approach.

## 7. Clinical Studies

The development of CECs injection therapy has been shaped by a robust body of preclinical research and a growing number of clinical trials, each contributing critical insights into the feasibility, safety, and efficacy of this regenerative approach. Early investigations in animal models laid the foundation for understanding how cultured CECs behave when introduced into the anterior chamber, while human studies have begun to validate the therapeutic potential in real-world settings [31,32]. An overview of the clinical trials for corneal endothelial cell injection therapy is presented in Table 8.

Vyznova (AURN001, Aurion Biotech, Seattle, WA, USA) is composed of allogeneic human corneal endothelial cells—known as “neltependocel”—combined with the ROCK inhibitor Y-27632. It represents the first approved corneal endothelial cell injection therapy in Japan. Clinical trials validated the therapeutic benefits seen in earlier studies, with over 80% of treated eyes demonstrating functional recovery and sustained endothelial integrity [11,31]. These promising outcomes led to regulatory approval in Japan in 2024, establishing CECs injection therapy as a viable alternative to keratoplasty for selected patients. Vyznova underwent clinical trials in the United States as part of its strategy for global market expansion. The Phase 1/2 CLARA trial (NCT06041256) assessed the safety, tolerability, and efficacy of AURN001 for 97 participants. The patients are assigned to receive either a low (2.5 × 10^5^), medium (5.0 × 10^5^), or high (1.0 × 10^6^) dose of neltependocel within the combination product. In an additional arm, participants receive neltependocel alone at a dose equivalent to that used in the high dose AURN001 arm and in yet another arm participants are treated with Y-27632 alone. The inclusion of Y-27632, a ROCK inhibitor, proved instrumental in enhancing cell adhesion and survival. According to the topline result pressed in December 2024, patients demonstrated significant improvements in visual acuity and corneal transparency within weeks of treatment, with endothelial cell density stabilizing over time. Importantly, no serious adverse events were reported, and the procedure was well tolerated, even in eyes previously considered high-risk for graft failure.

EO2002 (Emmecell, Menlo Park, CA, USA) developed a pioneering non-surgical cell therapy,) to treat corneal edema. It utilizes biocompatible magnetic nanoparticles integrated with cultured CECs to enhance targeted delivery and therapeutic efficacy. Topline results from a Phase 1, multi-center, randomized, double-masked clinical trial were released in November 2024. In the cohort receiving 150,000 cells, patients achieved a mean improvement of 11 letters in best-corrected visual acuity (BCVA) at 6 months, with 38% of subjects gaining at least 15 letters. All tested cohorts showed improvements in both BCVA and central corneal thickness, with no ocular or treatment-related serious adverse events reported.

Parallel efforts have explored the use of autologous iPSC-derived CECs, aiming to eliminate donor dependency and reduce immunogenic risk. Preliminary results from phase I trials suggest that these cells can be safely delivered into the anterior chamber without tumor formation or immune rejection [32]. While visual outcomes have been modest thus far, ongoing refinements in differentiation protocols and delivery techniques are expected to enhance efficacy in future cohorts.

Together, these studies underscore the translational momentum of CECs injection therapy. From proof-of-concept experiments in animal models to rigorously controlled human trials, the evidence base continues to expand, offering compelling support for the integration of cell-based approaches into mainstream ophthalmic practice. As clinical experience grows, long-term follow-up data will be essential to assess durability, monitor for late-onset complications, and refine patient selection criteria.

## 8. Challenges and Future Directions

Despite the encouraging progress in CECs injection therapy, several challenges remain that must be addressed before this approach can be widely adopted in clinical practice. One of the most pressing concerns is immunogenicity. Although the cornea is considered an immune-privileged site, allogeneic CECs may still elicit immune responses, particularly in eyes with prior inflammation or surgical history [162]. While the use of autologous iPSC-derived CECs offers a potential solution, these approaches are technically demanding and costly, and they require rigorous safety validation to eliminate residual pluripotent cells.

Standardization of cell culture protocols is another critical hurdle. Variability in donor tissue quality, media composition, substrate properties, and expansion techniques can lead to inconsistent cell phenotypes and therapeutic outcomes. Batch-to-batch variability can compromise reproducibility and regulatory approval. To address these issues, it is essential to establish Good Manufacturing Practice (GMP)-compliant workflows and implement validated markers for endothelial identity and function. Functional validation assays play a critical role in confirming therapeutic potential. Techniques such as TEER and Ussing chamber analysis can assess barrier integrity and ion transport, while organ-on-chip models may provide dynamic platforms that mimic physiological conditions and help predict in vivo behavior. Moreover, long-term safety data are still limited. While short-term results have demonstrated restored corneal clarity and improved vision, questions remain about the durability of engraftment, potential for EnMT, and risk of late-onset complications.

Scalability and cost-effectiveness also pose significant barriers. Producing therapeutic-grade CECs requires specialized facilities, trained personnel, and stringent quality control, all of which contribute to high production costs. Innovations in automation, cryopreservation, and centralized manufacturing may help reduce expenses and expand access, particularly in resource-limited settings. Additionally, the logistics of cell transport and storage must be optimized to maintain viability and potency during distribution.

Regulatory pathways for cell-based therapies are complex and evolving, with distinct frameworks across major regulatory regions. In the U.S., corneal endothelial cells (CECs) therapies are regulated under 21 CFR Part 1271 for Human Cells, Tissues, and Cellular and Tissue-Based Products (HCT/Ps). Products requiring substantial manipulation or non-homologous use fall under the Biologics License Application (BLA) pathway. Although accelerated programs such as the Regenerative Medicine Advanced Therapy (RMAT) designation can facilitate translation, these pathways still require rigorous long-term safety and efficacy data. In the European Union, corneal cell therapies are classified as Advanced Therapy Medicinal Products (ATMPs), regulated under a centralized procedure. The European Medicines Agency (EMA) offers early scientific advice to support the development of innovative regenerative therapies; however, harmonized GMP manufacturing standards and post-market surveillance are required to maintain authorization. In Japan, the Pharmaceuticals and Medical Devices Agency (PMDA) provides a conditional and time-limited approval system under the Act on the Safety of Regenerative Medicine. This framework enables earlier patient access through post-marketing data collection and has accelerated regenerative product approvals but also raises concerns regarding the long-term monitoring of efficacy and safety. Despite these regional differences, common challenges persist across regulatory frameworks. These include harmonization gaps in manufacturing and potency testing requirements, limited guidance on allogeneic versus autologous CEC therapy pathways, and the lack of standardized potency assays to assess corneal endothelial function. Additionally, ethical and logistical considerations surrounding the use of stem cell–based and gene-edited products further complicate regulatory navigation. Proactive engagement with regulatory agencies and transparent reporting of clinical outcomes will be essential to address these challenges. Moreover, there is an increasing need for globally harmonized regulatory frameworks capable of accommodating the unique biological and logistical characteristics of corneal endothelial cell therapies, which would ultimately accelerate clinical translation and broaden patient access.

Ensuring global equity remains a significant challenge in the advancement of regenerative ophthalmology. In low-resource regions, access to cutting-edge cell therapies is often constrained by limited infrastructure, high costs, and insufficient regulatory capacity. To overcome these barriers, decentralized manufacturing, modular GMP facilities, and open-access protocols can help expand access. Global collaboration and regulatory harmonization are key to ensuring regenerative therapies benefit underserved populations.

Looking forward, the convergence of biomaterials, gene editing technologies, and AI-driven quality control promises to elevate the precision and scalability of corneal endothelial cell (CEC) injection therapy. To translate laboratory breakthroughs into accessible, effective treatments for corneal blindness, sustained collaboration among academia, industry, and regulatory bodies will be crucial. Recent innovations in genetic and epigenetic modulation—such as CRISPR-based editing and histone modification—offer powerful tools to optimize CECs phenotypes, delay cellular senescence, and enhance regenerative capacity. Complementary approaches like exosome-based therapies can further improve cell survival and mitigate inflammation. Meanwhile, advanced bioengineering platforms—including microfluidics, 3D bioprinting, and biomimetic scaffolds—enable precise control over CECs delivery and integration. Together, these technologies address current limitations and accelerate the clinical translation of CECs injection therapy into mainstream ophthalmic care.

## 9. Conclusions

Corneal endothelial cell injection therapy represents a transformative advancement in the treatment of endothelial dysfunction, offering a minimally invasive, donor-independent alternative to traditional keratoplasty. Through innovations in cell sourcing, culture optimization, and delivery techniques, researchers have demonstrated the feasibility of restoring corneal clarity and function using bioengineered CECs. Clinical trials have validated the safety and efficacy of this approach, particularly when supported by ROCK inhibitors and precise postoperative management.

While challenges remain—including immunogenicity, standardization, scalability, and regulatory complexity—the field is rapidly evolving. Continued interdisciplinary collaboration and investment in translational research will be key to overcoming these barriers and realizing the full potential of cell-based therapies. As the global burden of corneal blindness continues to rise, CECs injection therapy offers a promising path forward—one that could redefine the future of ophthalmic care.

## Figures and Tables

**Figure 1 bioengineering-12-01162-f001:**
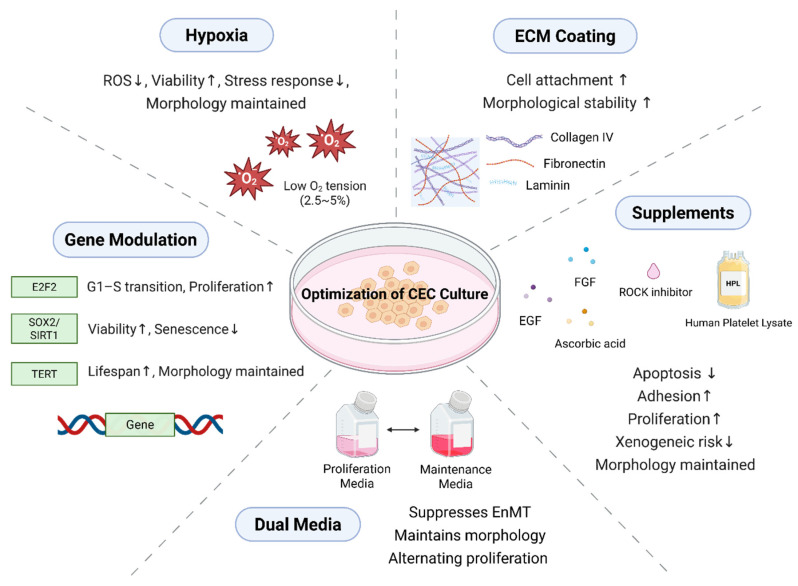
Optimization strategies for corneal endothelial cells (CECs) culture. Multiple factors enhance CECs proliferation and phenotype maintenance. Hypoxic culture (2.5–5% O_2_) mimics physiological oxygen levels, reducing reactive oxygen species and endoplasmic reticulum (ER) stress while improving viability. Extracellular matrix (ECM) coatings (collagen IV, laminin, fibronectin) promote adhesion and morphological stability. Supplements including ROCK inhibitor (Y-27632), FGF, EGF, ascorbic acid, and human platelet lysate (HPL) enhance proliferation and reduce apoptosis. Alternating proliferation and maintenance media preserves hexagonal morphology and suppresses endothelial-to-mesenchymal transition (EnMT). Gene modulation (E2F2, SOX2, SIRT1, TERT) promotes proliferation, delays senescence, and extends replicative lifespan.

**Figure 2 bioengineering-12-01162-f002:**
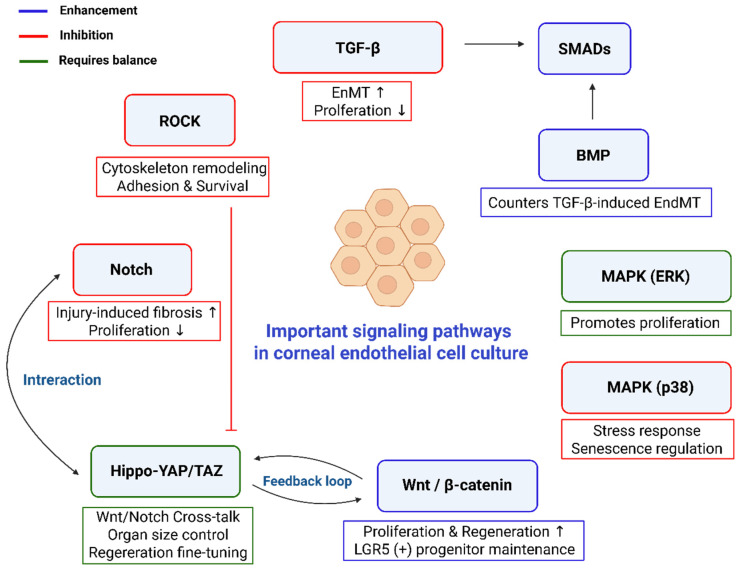
Signaling pathways maintain corneal endothelial cells (CECs) phenotype. Multiple signaling pathways—including TGF-β, BMP, MAPK (ERK and p38), Wnt/β-catenin, Hippo-YAP/TAZ, Notch, and ROCK- interaction to control CECs phenotype. TGF-β signaling promotes endothelial-mesenchymal transition (EnMT) and suppresses proliferation, while BMP counteracts TGF-β effects. Wnt/β-catenin and Hippo-YAP/TAZ pathways regulate regeneration and progenitor maintenance through feedback loops and crosstalk with Notch. ROCK supports cytoskeletal remodeling and adhesion, and MAPK signaling promotes proliferation stress responses. Coordinated regulation of these pathways is essential for maintaining CECs phenotype and homeostasis. Signal pathways highlighted in the blue box generally promote the phenotype and proliferation of CECs, whereas those in the red and green boxes represent inhibitory or controversial pathways.

**Figure 3 bioengineering-12-01162-f003:**
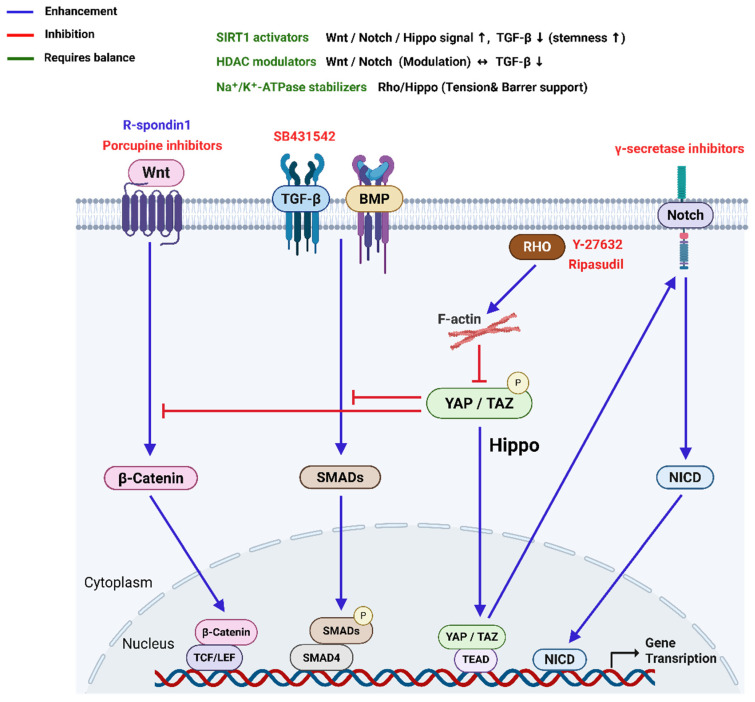
Crosstalk among signaling pathways regulating CECs phenotype. Diagram illustrating Wnt/β-catenin, TGF-β/BMP-SMAD, Notch, and Hippo-YAP/TAZ pathways. Wnt and Notch promote nuclear translocation of β-catenin and NICD, respectively, driving gene transcription. TGF-β/BMP activate SMADs and inhibit YAP/TAZ, while RHO-mediated F-actin dynamics enhance YAP/TAZ activity. Hippo signaling suppresses YAP/TAZ via phosphorylation. The pathways interact to fine-tune corneal endothelial cell behavior and phenotype maintenance.

**Figure 4 bioengineering-12-01162-f004:**
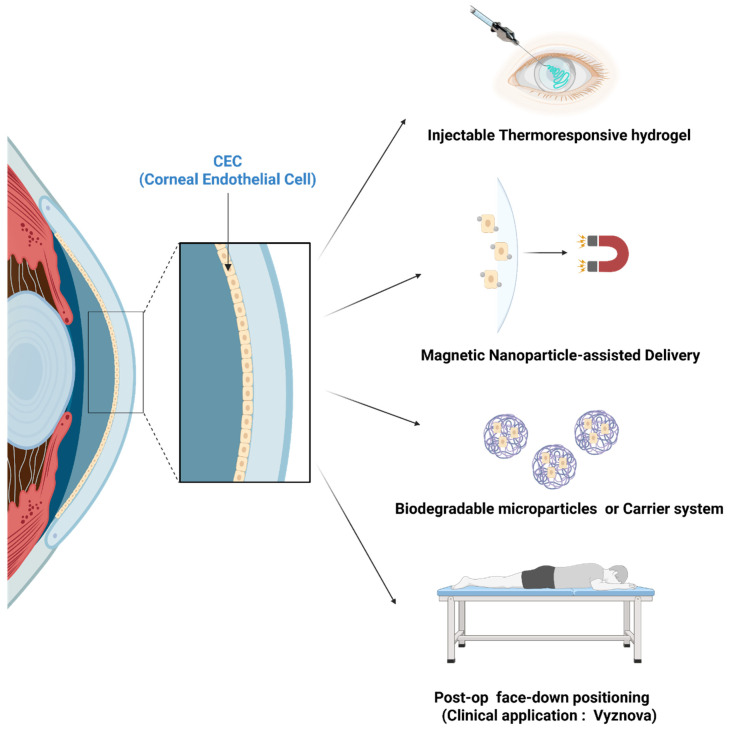
Delivery and Engraftment Approaches for CECs Therapy. Thermoresponsive and Hyaluronic acid (HA)-based hydrogels provide localized scaffolds, while magnetic nanoparticles guide CECs to Descemet’s membrane. Biodegradable carriers enable sustained release, and postoperative face-down positioning improves attachment—collectively supporting clinical translation exemplified by Vyznova therapy.

**Table 3 bioengineering-12-01162-t003:** Donor factors affecting successful corneal endothelial cell culture.

Factor	Specific Details & Findings	Influence on Culture Success	Reference
Donor Age	Optimal age range typically under 60 years; younger donors (≤50 years) tend to have higher endothelial cell viability and proliferative capacity.	Significantly higher success rates with younger donors.	[41]
Donor Health Status	Good systemic health, absence of diabetes, hypertension, ocular diseases like Fuchs endothelial dystrophy, or previous ocular surgeries.	Better endothelial cell quality and culture outcomes.	[61]
Cause of Death	Non-traumatic causes (e.g., natural causes) linked with better cell quality; traumatic deaths may cause cell damage.	Traumatic death reduces culturability and cell viability.	[61,62]
Death-to-Preservation Interval	Recommendations suggest within 6–12 h; shorter intervals improve cell viability.	Longer intervals (>12 h) negatively impact success.	[60]
Preservation Medium and Conditions	Use of preservative media like Optisol-GS, stored at 4 °C; delays in preservation decrease cell viability.	Proper preservation is critical to successful culture.	[63]
Endothelial Cell Density	Higher initial cell density (preferably > 2500 cells/mm^2^) correlates with better proliferation potential.	Low initial density (<2000 cells/mm^2^) reduces success.	[64]
Donor Sex	No consistent evidence suggesting sex significantly impacts endothelial cell culture outcomes.	Typically negligible effect.	[64,65]
Donor Age-Related Changes	Age-related decrease in cell proliferative capacity and wound healing ability.	Older donors (>60 years) have lower success rates.	[64,66]
Prior Ocular Surgeries or Treatments	Previous surgeries (e.g., cataract surgery) may induce endothelial cell loss or damage.	May decrease regenerative capacity, reducing success.	[67]

**Table 7 bioengineering-12-01162-t007:** Viral oncogenes to enhance the proliferation of human corneal endothelial cells.

Viral Oncogene	Details
SV40 Large T-antigen	Stimulates cell proliferation by inhibiting p53 and disrupting the Rb-E2F complex. Increases expression of CDK1, CDK2, and CDK4, and upregulates cyclin A and D [118]. Decreases cell cycle inhibitors p27KIP1 and p21CIP1. Results in increased proliferation rate and extended survival of CECs [128].
SV40 Small T-antigen	Increases cell proliferation by binding with protein phosphatase 2A and inhibiting heterochromatin protein 1-binding protein 3 [129]. Contributes to SV40 large T-antigen-mediated cell transformation. Expression of both large and small T-antigens results in similar proliferation effect on CECs as large T-antigen alone [130].
HPV-16 E6/E7	E6 oncoprotein increases cell proliferation by degrading p53 [131]. E7 induces ubiquitination of Rb proteins [132]. Stable expression of both E6 and E7 results in immortalization of CECs. Cells exhibit cobblestone-like polygonal morphology and are mostly diploid [133].

**Table 8 bioengineering-12-01162-t008:** Summary of clinical trials evaluating corneal endothelial cell injection therapy.

Trial	Goal	Design	Product Used	Primary Outcome Measure	Study Start/Completion	Results
CLARA trial (NCT06041256)	To compare different doses of AURN001 in patients with cornea edema secondary to corneal endothelial dysfunction	Phase 1/2, multicenter, randomized, double masked, prospective, parallel arm study	AURN001: combination of CEC and Y27632	BCVA 15 letters (3-lines) or more improvement at 6 months	18 October 2023/25 October 2024	The high-dose AURN001 group achieved the primary endpoint in 50% of participants, compared to just 14.3% in the group treated with Y27632 alone
EMME-001 (NCT04894110)	To evaluate the safety and tolerability of 3 doses of EO2002 with or without endothelial brushing or Descemet stripping in cornea edema secondary to corneal endothelial dysfunction with	Group 1: Phase 1, prospective, multi-center, open-label, dose escalation study Group 2: prospective, multi-center, double-masked study	EO2002: magnetic human corneal endothelial cells	Incidence of treatment-emergent adverse events at 26 weeks	22 June 2021/3 October 2024	Patients receiving 150,000 cells showed a mean 11-letter BCVA gain at six months, with 38% improving by at least 15 letters. All cohorts improved in BCVA and central corneal thickness
A first-in-human clinical study (doi.org/10.1016/j.xcrm.2024.101847)	To evaluate the safety and efficacy of an iPSC-derived corneal endothelial cell substitute.	Phase 1: first human trial for one patient with severe bullous keratopathy recurring after prior corneal transplantation	CLS001: iPSC derived corneal endothelial cells	Any adverse event	1 August 2022/30 March 2023	Over a 52-week follow-up, no tumor formation or severe inflammation was observed, and patients showed improved visual acuity with a trend toward restored corneal transparency

## Data Availability

The original contributions presented in this study are included in the article. Further inquiries can be directed to the corresponding author.

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
