# Peer review of "Bioengineering Strategies for Corneal Endothelial Cell Injection Therapy: Advances, Challenges, and Clinical Translation"

_bioengineering, 2025, doi:10.3390/bioengineering12111162_

Round 1

Reviewer 1 Report

Comments and Suggestions for Authors

The authors presented manuscript entitled Bioengineering Strategies for Corneal Endothelial Cell Injection Therapy: Advances, Challenges, and Clinical Translation can be further considered on improvement. These suggestions aim to enhance the comprehensiveness, clarity, and translational relevance of this valuable review on corneal endothelial cell injection therapy 

  1. Suggested to add summary tables or schematic figures to visualize complex topics such as signaling pathways influencing CEC phenotype, key bioengineering strategies, and clinical trial outcomes to improve reader comprehension.
  2. Suggested to expand discussion on alternative cell sources beyond just iPSCs and ESCs, including trans-differentiated somatic cells and non-corneal sources, to provide a comprehensive evaluation of options and their translational potential.
  3. Suggested to include more critical analysis and comparison of culture media compositions and supplements, highlighting advantages and limitations of human serum, platelet lysate, serum-free options, and commonly used growth factors and inhibitors.
  4. Suggested to elaborate on the challenges and current solutions in manufacturing and quality control, including Good Manufacturing Practice (GMP) considerations, batch-to-batch variability, and functional validation assays such as TEER, Ussing chamber, and organ-on-chip models.
  5. Suggested to provide more detailed discussion on the hurdles in clinical translation, specifically immunogenicity risks, standardization of protocols, scalability of cell production, and long-term safety and efficacy data.
  6. Suggested to discuss regulatory landscapes across different regions more explicitly, addressing how existing frameworks accommodate innovative cell therapy products and highlighting gaps/challenges.
  7. Possibly include updates on the latest clinical trials and approved therapies such as Vyznova, EO2002, and others, with more detail on clinical endpoints, patient outcomes, and adverse events to contextualize therapeutic progress.
  8. Discuss the potential impact and current state of biomaterial-based delivery systems like hydrogels, magnetic nanoparticles, and biodegradable carriers, along with their mechanistic advantages and translational hurdles.
  9. Mention the role of genetic and epigenetic modulation, exosomes, and advanced bioengineering tools, emphasizing how they can further improve CEC viability, function, and engraftment.
  10. Ensure consistent and clear use of abbreviations and terminology throughout the manuscript, with a glossary if necessary.
Comments on the Quality of English Language
  1. Please use more active voice and concise sentences where possible, especially in the Results and Discussion sections.

Reviewer 2 Report

Comments and Suggestions for Authors
  1. The introduction repeats some concepts (e.g., limitations of keratoplasty) from later sections—streamline to avoid redundancy and improve flow.
  2. The Authors should expand on global disparities in donor availability with more recent data (post-2020) to reflect current statistics, as the cited estimates (e.g., 1:70 ratio) may need updating for 2025 relevance. Include a brief comparison table of PK, DSAEK, and DMEK outcomes for clarity.
  3. Table 1 could include dosage ranges and potential side effects for each supplement to enhance practical utility.
  4. Table 2 is useful, but the authors should incorporate gender-specific data more robustly, as the current note of "negligible effect" could be expanded with recent studies on sex differences in cell viability.
  5. Figures 2 and 3 are informative, but the authors should add arrows or labels for crosstalk interactions to make them more intuitive and discuss potential therapeutic targets (e.g., small molecules for Wnt/Notch) in greater depth for forward-looking insights.
  6. Table 4 lists markers well, but I recommend including flow cytometry thresholds or validation methods (e.g., qPCR primers) to aid reproducibility. Address how these markers correlate with in vivo function more explicitly.
  7. The organ-on-a-chip discussion is forward-thinking, but cite specific examples of cornea-on-chip models from 2024-2025 literature to strengthen this section. The authors should add a protocol flowchart for TEER or using chamber assays.
  8. Tables 5 and 6 cover viral oncogenes, but the authors should emphasize on safety concerns like tumorigenicity more prominently, perhaps with a risk-benefit analysis. Update with CRISPR-based alternatives for non-viral approaches.
  9. The magnetic nanoparticle discussion is exciting, but the authors should include limitations such as potential toxicity or regulatory hurdles. Add a figure comparing gravity vs. magnetic delivery efficiencies.
  10. Good coverage of iPSCs and ESCs, but add a paragraph on ethical considerations for ESCs and cost comparisons with donor-derived cells. Validate animal model translations to humans with more critical analysis.
  11. Figure 4 is helpful, but label components more clearly (e.g., specific hydrogels). Expand on postoperative complications like IOP spikes and mitigation strategies.
  12. In the challenges and direction section, the authors should propose specific solutions, like AI for quality control or blockchain for donor tracking. Add global equity discussions for low-resource regions.
  13. Proofread for minor grammatical issues (e.g., inconsistent superscript references, awkward phrasing like "pressed in 2024").

Round 2

Reviewer 1 Report

Comments and Suggestions for Authors

The authors have reflected all the said suggestions and comments, which made the manuscript enhanced with improved readability; Thus, I suggest for further consideration with acceptance.

Reviewer 2 Report

Comments and Suggestions for Authors

The authors have carefully addressed my raised concerns.